# Isolation and characterization of novel plasmid-dependent phages infecting bacteria carrying diverse conjugative plasmids

Boris Parra,[1,2,3] Bastiaan Cockx,[1] Veronika T. Lutz,[4] Lone Brøndsted,[4] Barth F. Smets,[1] Arnaud Dechesne[1]

**ABSTRACT** Plasmid-dependent phages infect bacteria carrying conjugative plasmids by recognizing the plasmid-encoded pilus. Despite the high abundance of conjugative plasmids in diverse environments, plasmid-dependent phages have not been widely studied. Since conjugative plasmids often carry antimicrobial resistance genes (ARGs), interfering with conjugation could reduce the spread of ARGs and avoid the appearance of multiresistant superbugs. Our aim was to isolate and characterize plasmid-dependent phages able to infect bacteria carrying diverse conjugative plasmids belonging to the most common plasmid families among Gram-negative pathogens. We isolated two lytic phages from wastewater using an avirulent strain of *Salmonella enterica* carrying the conjugative IncN plasmid pKM101. Both phages, named Lu221 and Hi226, are novel dsDNA viruses within the class *Caudoviricetes* with genomes of approximately 76 kb. They showed broad host range infecting *Escherichia coli*, *S. enterica*, *Kluyvera* sp., and *Enterobacter* sp. carrying conjugative plasmids. They recognize plasmid-encoded receptors from 12 out of 15 tested plasmids, all of them carrying resistance determinants. Phages Lu221 and Hi226 could have the potential to help combat the antimicrobial resistance crisis by reducing ARGs present in conjugative plasmids.

**IMPORTANCE** This work was undertaken because plasmid-dependent phages can reduce the prevalence of conjugative plasmids and can be leveraged to prevent the acquisition and dissemination of ARGs by bacteria. The two novel phages described in this study, Lu221 and Hi226, can infect *Escherichia coli*, *Salmonella enterica*, *Kluyvera* sp. and *Enterobacter* sp. carrying conjugative plasmids. This was verified with plasmids carrying resistance determinants and belonging to the most common plasmid families among Gram-negative pathogens. Therefore, the newly isolated phages could have the potential to help control the spread of ARGs and thus help combat the antimicrobial resistance crisis.

**KEYWORDS** bacteriophages, pilus, wastewater, *Caudoviricetes*, horizontal gene transfer, conjugation, antimicrobial resistance

Conjugation is the most frequent mechanism of horizontal gene transfer in bacteria (1, 2), and among the thousands of known plasmids, one-fourth are conjugative (3). Since antimicrobial resistance genes (ARGs) are frequently carried by conjugative plasmids (4, 5), control of conjugation has been proposed as a strategy to combat antimicrobial resistance crisis (AMR) (6–8). Such control can be achieved by inhibiting conjugation and/or by promoting plasmid loss (9–11). In this context, plasmid-dependent phages have been proposed to control the spread of ARG, because they are able to infect and subsequently kill bacteria carrying conjugative plasmids (12–14). These phages initiate infection by attaching to a plasmid-encoded receptor, and, therefore, they are also known as male-specific or plasmid-specific phages (15, 16).

Address correspondence to Arnaud Dechesne, arde@dtu.dk.

The authors declare no conflict of interest.

See the funding table on p. 12.

*[This article was published on 8 December 2023 with incorrect burst size values in the "Adsorption assay, one-step growth curve, and growth inhibition assay" section and in Table 3. These values were corrected in the current version, posted on 22 December 2023.]*

Most plasmid-dependent phages were isolated from wastewater several decades ago (17–19) but have received little attention since then, except for few coliphages such as MS2 and M13, which were instrumental to early molecular biology and genetics (20, 21). Even so, the knowledge of these phages is limited, although some recent studies have shown that some of them can have a great potential to be used to reduce ARGs' load in microbial communities (22–24). Recently, He et al. (25) and Quinones-Olvera et al. (26) demonstrated that plasmid-dependent phages are in fact common and abundant in diverse environments.

Our aim was to isolate and characterize plasmid-dependent phages infecting bacteria carrying conjugative plasmids that confer resistance against clinically important ARGs that are also found in the environment. In this work, we describe lytic phages Lu221 and Hi226, isolated using the IncN plasmid pKM101, a conjugative multidrug-resistant plasmid that confers resistance against ampicillin, streptomycin, sulfonamide, and tetracycline (27). Moreover, we studied their host range (i.e., infectiveness against diverse bacteria carrying diverse plasmids) and analyzed their genomes. We confirmed that they are unique phages belonging to a novel genus in the *Caudoviricetes* class, according to the current classification proposed by the International Code of Virus Classification and Nomenclature (ICTV) (28).

## RESULTS AND DISCUSSION

### Isolation of plasmid-dependent phages

To isolate plasmid-dependent phages from wastewater and distinguish them from somatic phages (i.e. phages whose infection is not plasmid-dependent), we used conventional double-layer agar (DLA) method with two strains of *Salmonella enterica* MHM112 (29): one strain without plasmids and another carrying the conjugative plasmid pKM101. Wastewater samples were obtained from the influent of two Danish wastewater treatment plants (WWTPs) located in Lundtofte and Hillerød.

Overall, the number of plaques detected with the plasmid-carrying strain suggests that the wastewater contained lytic phages at a density of about $10^4$ PFU mL$^{-1}$. This estimate combined the abundance of pKM101-dependent phages and of *S. enterica* somatic phages. This abundance is much larger than that estimated from the plates with the strain without plasmid, indicating that plasmid-dependent phages were more abundant than *S. enterica* somatic phages in both WWTPs (Table 1). This dominance of plasmid-dependent phages is consistent with our previous work quantifying phages at the same locations (25).

We then proceeded to isolate phages from eight plaques for each sample. They were purified and tested against the strain without plasmid, confirming that they are all plasmid-dependent. Due to their similar phenotypic characteristics such as plaque morphology, RNAse and chloroform resistance, we selected one phage from each sampling location for further characterization and sequencing: phage Lu221 from Lundtofte WWTP and phage Hi226 from Hillerød WWTP.

### Characterization of phages Lu221 and Hi226

Transmission electron microscopy revealed that both phages present the same morphology, with an elongated head and a short tail (Fig. 1). The head of phage Lu221 has a length of 152 ± 10 nm and a width of 69 ± 8 nm, while phage Lu221 has a head

TABLE 1   Abundance of phages counted by DLA from wastewater samples using *S. enterica* MHM112 (pKM101)[a]

| | Somatic | | | IncN-dependent | | |
|---|---|---|---|---|---|---|
| | Total | DNA | RNA | Total | DNA | RNA |
| Lundtofte | $1.5 \times 10^2 \pm 7.1 \times 10^1$ | 0 | $1.5 \times 10^2 \pm 7.1 \times 10^1$ | $3.8 \times 10^4 \pm 2.1 \times 10^3$ | $3.2 \times 10^4 \pm 3.5 \times 10^3$ | $6.7 \times 10^3 \pm 5.7 \times 10^3$ |
| Hillerød | $5.0 \times 10^1 \pm 7.1 \times 10^1$ | $5.0 \times 10^1 \pm 7.1 \times 10^1$ | 0 | $4.5 \times 10^4 \pm 3.0 \times 10^3$ | $4.2 \times 10^4 \pm 3.5 \times 10^3$ | $3.7 \times 10^3 \pm 2.1 \times 10^2$ |

[a]Results are presented as mean values (PFU mL$^{-1}$) from two independent replicates.

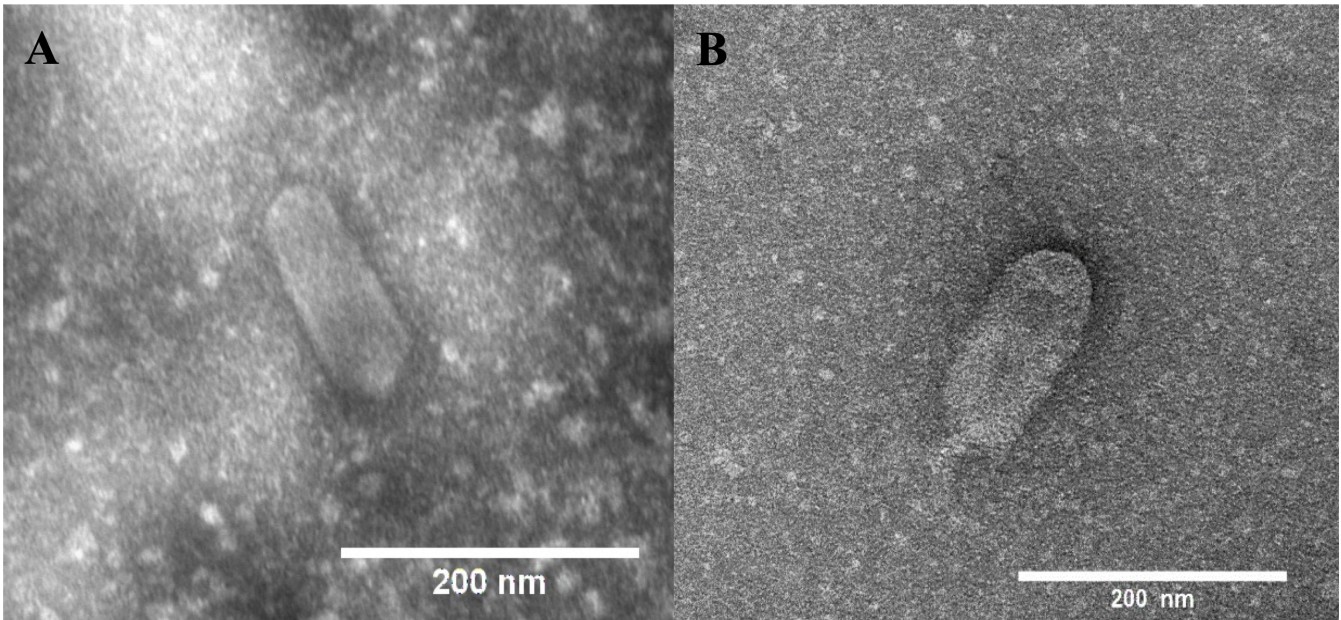

**FIG 1** Transmission electron micrograph of phages Lu221 and Hi226 (A and B, respectively) showing their elongated heads and short tails. Scale bars = 200 nm.

length of 151 ± 10 nm and a width of 70 ± 9 nm. The prolate head, also refered to as a C3 morphotype, is an unusual characteristic among phages (30).

To the best of our knowledge, no enterobacteria podophages have been described as plasmid-dependent. However, most descriptions of such phages have not determined whether they are somatic or plasmid-dependent, which is possible since many were isolated using bacteria that could harbor conjugative plasmids, such as *S. enterica*, *Escherichia coli*, *Kluyvera* sp., etc. (31, 32). On the other hand, podovirus phage MPK7 infecting *Pseudomonas aeruginosa* PAO1 requires type IV pili for infection (33), which is encoded on an IncP conjugative plasmid (34). Moreover, two other podophages that require type IV pili for infection have been described, able to infect both phytopathogens *Xylella fastidiosa* and *Xanthomonas* sp., phages Prado and Paz (35).

## Genome analysis of phages Lu221 and Hi226

The genomes of phages Lu221 and Hi226 are dsDNA molecules of 76.397 bp and 76.506 bp, respectively, with a GC content of 43.5% and nucleotide identity of 95.7% (Fig. 2A). Their closely related phages are pSal-SNUABM-01, 7–11, and SE131, with ~78% identity to our newly isolated phages. Those phages were also isolated using a *Salmonella* strain as the host bacterium and belong to the newly proposed family *Grimontviridae* in the *Caudoviricetes* class (36). Phage pSal-SNUABM-01 was isolated in South Korea and has an 89.5-kb genome with a GC content of 45.4% (37), phage 7–11 was isolated in Canada and has an 89.9-kb genome with a GC content of 44.1% (38, 39), and phage SE131 was isolated in South Korea and has an 89.9-kb genome with a GC content of 44.2%. Thus, phages Lu221 and Hi226 have lower GC content and shorter genomes compared to their close relatives (Fig. 2B). The difference in gene content consists of genes encoding hypothetical proteins with unknown functions. Phages pSal-SNUABM-01, 7–11, and SE131 were classified in the former *Podoviridae* family; however, drastic changes in phage classification were recently officialized by ICTV, including the abolishment of the morphology-based family *Podoviridae* and order *Caudovirales* (28). Therefore, they are now considered unclassified phages in terms of order but belong to the recently proposed family *Grimontviridae* in the class *Caudoviricetes* (36).

**TABLE 2** Phages related to Hi226 and Lu221 belonging to the recently proposed new family *Grimontviridae* (36)

| Phage | Bacterial host | Genome size | Accession | Country | Reference |
|---|---|---|---|---|---|
| pSal-SNUABM-01 | *Salmonella* | 89,500 | MW296032.1 | South Korea | (37) |
| 7–11 | *Salmonella* | 89,916 | HM997019.1 | Canada | (42) |
| SE131 | *Salmonella* | 89,910 | MG873442.1 (NC_070974.1) | South Korea | -- |
| vB_CsaP_GAP52 | *Cronobacter* | 76,631 | JN882286.1 | Canada | -- |
| vB_CtuP_A24 | *Cronobacter* | 75,106 | MW343794.1 | China | (43) |
| BUCT695 | *Aeromonas* | 86,289 | OL799327.1 | China | -- |
| vB_CsaP_009 | *Proteus* | 92,122 | NC_048664.1 | Iran | -- |
| Privateer | *Proteus* | 90,710 | NC_048861.1 | USA | (41) |
| Vb_PmiP-P59 | *Proteus* | 90,187 | MT664722.1 | China | -- |
| 3H10_20 | *Proteus* | 90,382 | MT740244.1 | Egypt | -- |
| 4_4572 | *Aeromonas* | 102,915 | NC_048773.1 | China | (44) |
| LAh_6 | *Aeromonas* | 101,390 | NC_048774.1 | Australia | (45) |
| LAh_8 | *Aeromonas* | 97,408 | NC_048775.1 | Australia | (45) |
| LAh_9 | *Aeromonas* | 97,988 | NC_048776.1 | Australia | (45) |

Other phages related to Lu221 and Hi226 are listed in Table 2. They include *Cronobacter* phages vB_CsaP_GAP52, A24, and vB_CsaP_009; *Aeromonas* phages BUCT695, LAh_6, LAh_8, LAh_9, and 4_4572; *Proteus* phages 3H10_20, Privateer, and vB_Csap_009; and *Kuravirus*-like phages, which are abundant *E. coli* viruses (40, 41).

The genome of phages Lu221 and Hi221 contains 133 ORFs, 28 (21.1%) of which were detected on the positive strand and 105 (78.9%) on the negative strand (Fig. 3). Only 39 ORFs (30.2%) were annotated as functional proteins, while the others were defined as hypothetical proteins with unknown functions. The functional ORFs were divided into four categories: structure- and packaging-related, nucleotide metabolism-related, lysis-related, and additional function (Table S2). In comparison, the genome of phage pSal-SNUABM-01 contains 162 ORFs (37), of which 94 are proteins homologous to Lu221. Collectively, all these homologous proteins present an average identity of 63.5%. The

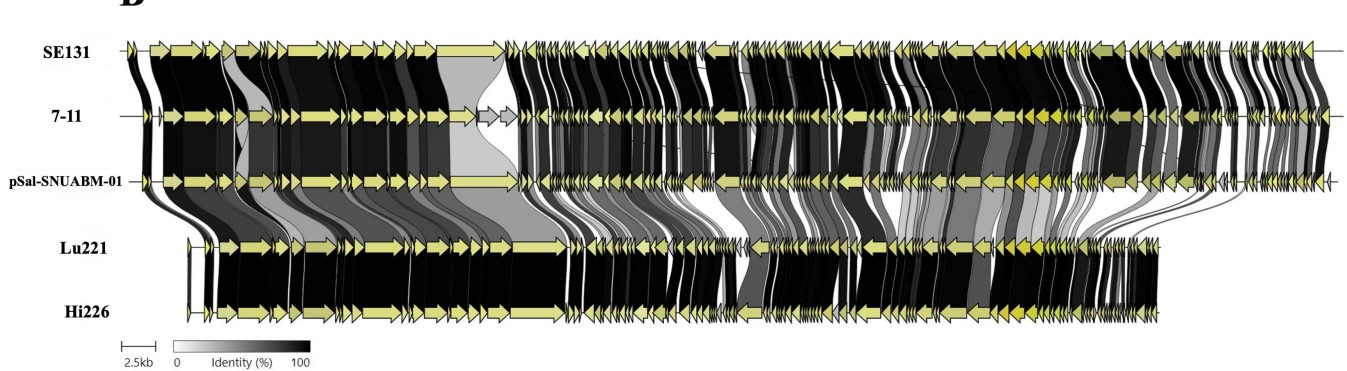

**A**

| | Hi226 | Lu221 | pSal-SNUABM-01 | 7-11 | SE131 |
|---|---|---|---|---|---|
| Hi226 | 100 | 95.7 | 78.9 | 78.7 | 78.5 |
| Lu221 | | 100 | 78.9 | 78.7 | 78.4 |
| pSal-SNUABM-01 | | | 100 | 90.5 | 82.6 |
| 7-11 | | | | 100 | 95.3 |
| SE131 | | | | | 100 |

**B**

**FIG 2** Comparative whole-genome analysis of phages Hi226 and Lu221 and the closely related phages pSal-SNUABM-01, 7–11, and SE131. Overall nucleic acid sequence identity between pairwise genomes determined by BLASTN at NCBI (A) and alignment of all annotated protein sequences obtained using Clinker software (B).

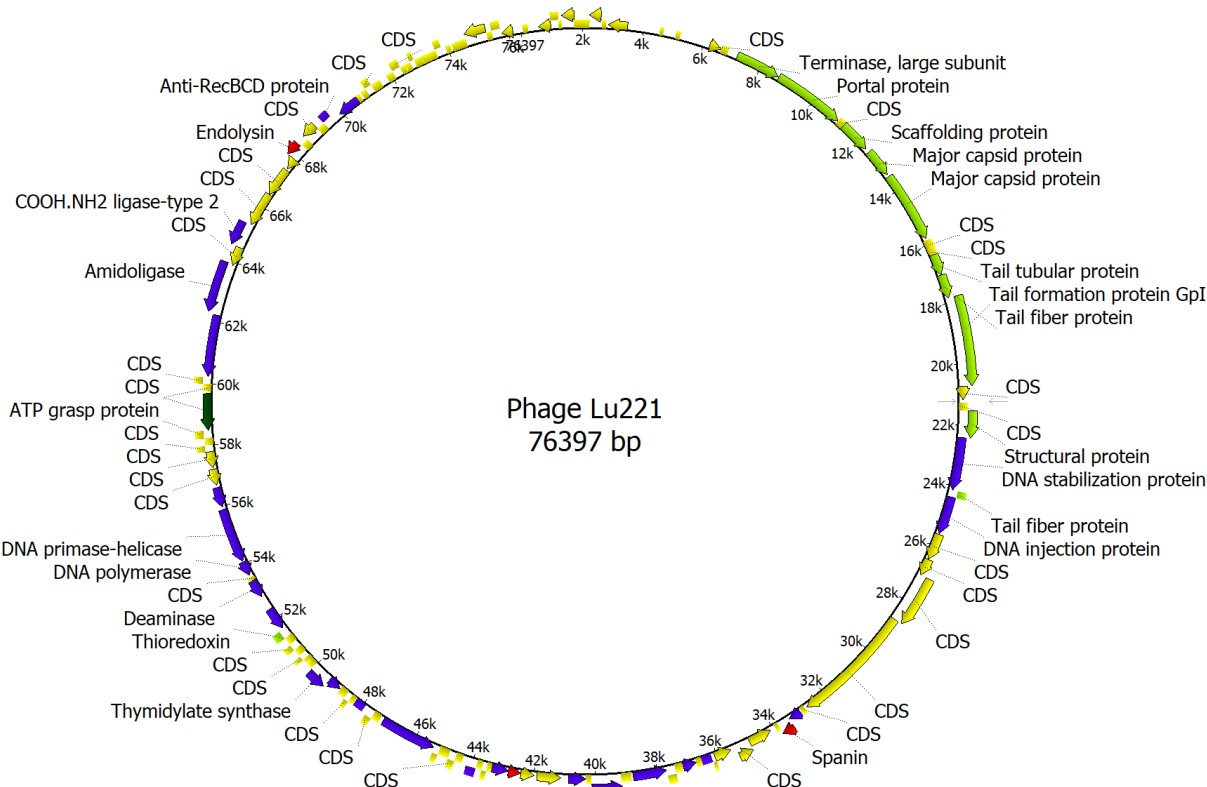

**FIG 3** Genome map of novel plasmid-dependent phage Lu221. The ORFs are shown in different colors to indicate their functional category. Red, lysis-related genes; green, structure- and packaging-related genes; blue, nucleotide metabolism-related genes; and yellow, hypothetical genes.

most conserved proteins are one of the two putative major capsid proteins (93.6% of identity), a hypothetical protein (92.5%) (QQK87877.1 in pSal-SNUABM-01), the putative terminase large subunit (91.7%), and the putative tail tubular protein (87.3%) (Table S2). These results are consistent with previous observations made by Mirzaei et al. (46), Kwon et al. (37), and Batinovic et al. (47), who suggested that the structural genes are the most conserved. On the other hand, no sequences homologous to known integrase, virulence factors, resistance genes, or any other harmful factors were identified in phages Lu221 and Hi221. Similarly, no tRNA genes were detected.

To explore the taxonomy of the two isolates, we downloaded the genomes of 33 phages from Genbank in March 2023, including pSal-SNUABM-01, 7–11, SE131; *Cronobacter* phages vB_CsaP_GAP52, A24, and vB_CsaP_009; *Aeromonas* phages BUCT695, LAh_6, LAh_8, LAh_9, and 4_4572; *Proteus* phages 3H10_20, Privateer, and vB_Csap_009; and 19 *Kuraviruses*-like phages (Fig. 4). According to our results, phages Lu221 and Hi226 belong to the same genus as pSal-SNUABM-01, 7–11, and SE131, which form the recently proposed genus *Moazamivirus* (36). In addition, they are related to the recently proposed genus *Crifsvirus*, which includes *Cronobacter* phages vB_CsaP_GAP52 and A24 (36); the recently proposed genus *Libingvirus*, which include only *Aeromonas* phage BUCT695 (36); *Privateervirus* genus, which was proposed by Adriaenssens et al. (48) but was recently modified to include *Proteus* phages Privateer, Vb_PmiP-P59 and 3H10_20 (36); and *Lahexavirus* genus proposed by Kabwe et al. (49), which include *Aeromonas* phages LAh_6, LAh_8, LAh_9, and 4_4572. These genera form the novel family *Grimontviridae* recently proposed by Moraru et al. (36), which, according to these authors, could be combined at the order level with the *Kuravirus*-like family.

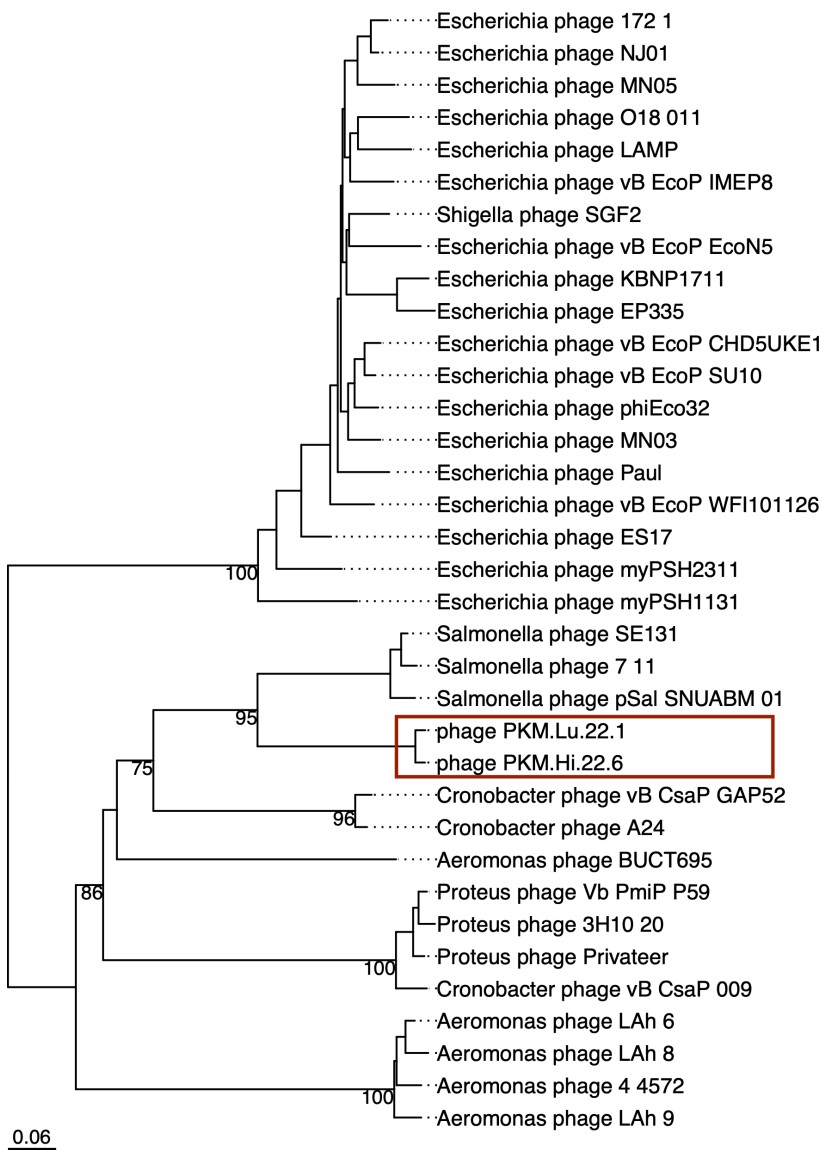

**FIG 4** Taxonomic assignment of phages Lu221 and Hi226 among related members in *Caudoviricetes*. Whole genome-based phylogeny was inferred using VICTOR (Virus Classification and Tree Building Online Resource), based on formula D0. the numbers above branches are the GBDP pseudo-bootstrap support values from 100 replications. The two phages isolated in this work are highligthed in a red box.

## Adsorption assay, one-step growth curve, and growth inhibition assay

The infection cycles of phages Lu221 and Hi226 were characterized by performing adsorption assays and one-step growth curves using *S. enterica* (pKM101). Phage Lu221 has an adsorption rate of approximately $3.5 \pm 1.1 \times 10^{-9}$ mL min$^{-1}$ and a burst size of 107 PFU/infected cell, while phage Hi226 presents similar characteristics with an adsorption rate constant of $5.9 \pm 1.9 \times 10^{-9}$ mL min$^{-1}$ and a burst size of 108 PFU/infected cell (Table 3). In addition, we demonstrated, in a growth inhibition assay, that the addition of phages Lu221 or Hi226 at a multiplicity of infection (MOI) of 10 prevented the growth of *S. enterica* (pKM101) (Fig. 5). In the presence of the phages, the bacterial cultures carrying the plasmid maintained their $OD_{600}$ similar to the beginning of the experiment after 210 min of incubation.

**TABLE 3** Phenotypic characteristics of phages Hi226 and Lu221[a]

| | Adsorption % | | | Adsorption rate | Burst size |
|---|---|---|---|---|---|
| | **15 min** | **30 min** | **45 min** | **constant (mL min$^{-1}$)** | **PFU/bacterial cells** |
| Phage Lu221 | 18.5 ± 1.2 | 34.2 ± 1.7 | 79.4 ± 5.8 | 3.5 ± 1.1 × 10$^{-9}$ | 107 |
| Phage Hi226 | 21.9 ± 4.8 | 32.1 ± 9.6 | 93.2 ± 0.5 | 5.9 ± 1.9 × 10$^{-9}$ | 108 |

[a]Results are presented as mean values ± SD from two independent experiments.

## Host range of phages Lu221 and Hi226

We tested the ability of phages Lu221 and Hi226 to infect diverse bacteria (Fig. 6). Our results revealed the broad host range of the isolated phages, because they infected plasmid-containing *E. coli*, *S. enterica*, *Kluyvera* sp., and *Enterobacter* sp. They were unable to infect *Pseudomonas putida*, which suggests a specificity toward *Enterobacteriaceae*. In this regard, other plasmid-dependent phages also showed a broad host range; for instance, PRD1 can infect *Salmonella*, *Pseudomonas*, *Escherichia*, *Proteus*, *Vibrio*, *Acinetobacter*, and *Serratia* strains carrying conjugative plasmids (12, 17). In terms of plasmid diversity, the two isolates are able to infect *Enterobacteriaceae* carrying 12 of the 15 plasmids carrying resistance determinants tested, spanning seven Inc families, IncH, IncI, IncN, IncP, IncW, and IncX. Another plasmid-dependent phage able to infect a similarly diverse range of plasmids is phage X, which is an ssDNA filamentous phage (18). However, this phage was not taxonomically classified, and its genome sequence is not available (15).

Our results demonstrated that phages Lu221 and Hi226 infect bacteria carrying conjugative plasmids R477-1 and pOXA436. This is the first description of DNA phages infecting bacteria carrying plasmids from the IncH group, as all other IncH-dependent phages are RNA viruses, which exclusively infect bacteria carrying IncH plasmids (50).

In some cases, different infection profiles were observed for different bacteria carrying the same plasmid using our isolated plasmid-dependent phages. For instance, *S. enterica* (RP4) was infected, but not *E. coli* (RP4). We tested the conjugation ability of *E. coli* (RP4), demonstrating that this strain can transfer the plasmid, and therefore the pilus is likely intact. Differences in infectivity of plasmid-dependent phages on different bacteria carrying the same plasmid have been described previously. This may be due to (i) different expression of mating pair formation (*mpf*) genes, (ii) a non-plasmid coded structure stabilizing phage attachment, or (iii) bacteria differing in their defense mechanisms against phages (antiphage systems). Expression of *mpf* genes of a same plasmid can vary across different bacterial hosts. Walker (51) demonstrated

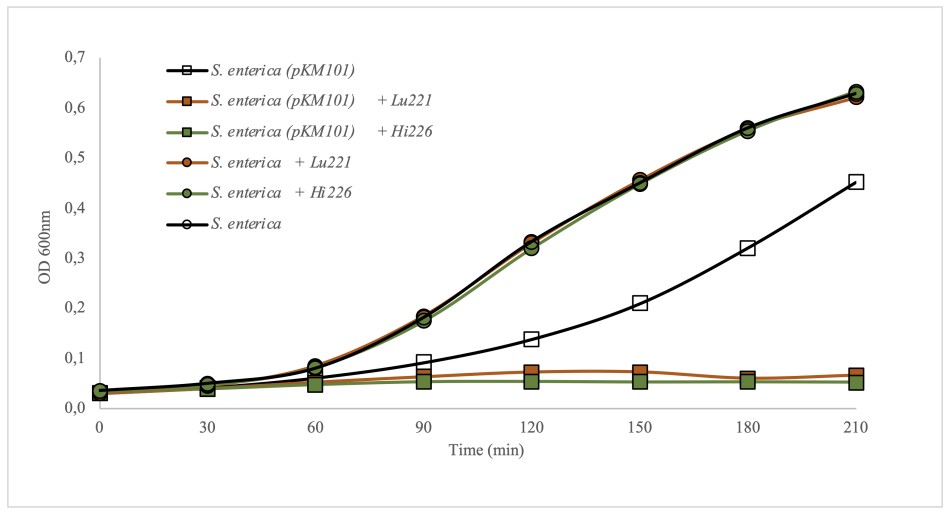

**FIG 5** Growth inhibition by phage Lu221 or Hi226. The phages were tested at an MOI of 10 against the host strain *S. enterica* (pKM101) and its plasmid-free counterpart (*S. enterica*). All experiments were performed in LB broth at 25°C.

| Bacterial host | Plasmid | Inc group | Hi226 | Lu221 |
|---|---|---|---|---|
| S. enterica MHM112 | drR27 | H | | |
| | pKM101 | N | | |
| | pKJK5 | P | | |
| | RP4 | | | |
| | drd19 | F | | |
| E. coli K12 | drR27 | H | | |
| | R477-1 | | | |
| | pOXA436 | | | |
| | R64 | I | | |
| | R721 | | | |
| | pTR4 | N | | |
| | RN3 | | | |
| | pKM101 | | | |
| | pB10 | P | | |
| | RP4 | | | |
| | pKJK5 | | | |
| | R388 | W | | |
| | pIE321 | | | |
| | R6K | X | | |
| Kluyvera sp. | drR27 | H | | |
| | pKM101 | N | | |
| | pKJK5 | P | | |
| | drd19 | F | | |
| Enterobacter sp. | pKM101 | N | | |
| | drR27 | H | | |
| P. putida KT2440 | RP4 | P | | |
| | pKJK5 | | | |
| | R388 | W | | |

FIG 6 Host range of phages Lu221 and Hi226 in terms of bacterial hosts carrying conjugative plasmids. Colors represent the phage infectivity, where green boxes indicate no plaques, yellow boxes percentile 50, and red boxes highest plaquing values. Even at higher phage concentrations, no clearing was observed in the lawn of negative strains. For details, please see Table S3.

that the transfer of plasmid pKM101 from *Salmonella typhimurium* to *E. coli* proceeded at higher efficiency than transfer in the reverse direction, while Kotilainen et al. (52) demonstrated that there are 50 PRD1 binding sites in *S. enterica* carrying plasmid RP1, whereas there are only 20 in *E. coli* with same plasmid. Therefore, in both cases, these differences may influence infection efficiency of plasmid-dependent phages. Although plasmid-dependent phages usually recognize a component of the mating pair formation system for the initial adsorption, other structures on the surface of the bacteria may stabilize phage attachment and allow effective infection (53). In fact, the pilus is a retractile appendage, and according to Skerker and Berg (54), this ability plays a role in the translocation of plasmid-dependent phages to the cell surface. Therefore, we cannot rule out that infection of Lu221 and Hi226, in addition to the pilus, depends on other structures that differ between the bacteria used in this work. Probably these have different antiphage systems, such as specific restriction/modification systems, toxin-anti-toxin systems associated with abortive infection or CRISPR-Cas (55, 56).

Phage 7–11 is one of the closest phages to Lu221 and Hi226. It was isolated using *Salmonella* Newport; but according to Grimont and Grimont (57), it is able to produce plaques on *E. coli*, *Salmonella* Panama, *Levinea malonatica*, *Enterobacter cloacae*, *Cronobacter sakazakii*, and *Erwinia herbicola*, while it is unable to produce plaques in

related bacteria such as *Shigella* spp., *Citrobacter freundii*, *Salmonella arizonae*, *Salmonella paratyphi*, and *Salmonella typhi*. The authors did not indicate the presence of plasmids in these bacterial strains, so it cannot be ruled out that phage 7–11 is a plasmid-dependent phage.

In conclusion, in this work, we describe two novel lytic plasmid-dependent phages isolated from wastewater, Lu221 and Hi226. They showed a broad host range in terms of bacterial strains and were able to recognize 12 conjugative plasmids carrying resistance determinants and belong to the most common plasmid families among Gram-negative pathogens. The characteristics of phages Lu221 and Hi226 make them potential agents to help combat the AMR crisis by reducing the spread of ARGs commonly present on conjugative plasmids.

## MATERIALS AND METHODS

### Bacterial strains and growth conditions

We used bacterial strain *S. enterica* MHM112, cured of the IncF-like plasmid, but with and without plasmid pKM101 (27) to isolate plasmid-dependent phages from wastewater samples (25). This strain was used because the abundance of *Salmonella* somatic phages in wastewater is low compared to other enterobacteria (58). In addition, several bacterial strains, carrying diverse plasmids from several Inc families, were used to determine the host range of isolated plasmid-dependent phages (Table S3). These bacterial strains were generated by mating assays and selection with antibiotics. Due to the lack of selectable markers in *S. enterica* and *E. coli* BL21, the plasmids were initially transferred to an auxotroph strain *E. coli* ST18, a ΔhemA deletion mutant requiring 5-aminolevulinic acid (ALA) for growth (59) and resistant to streptomycin. In mating experiment, donor and recipient strains were grown overnight in Luria-Bertani (LB) media with appropriate antibiotics and ALA 50 µg mL$^{-1}$ (for *E. coli* ST18) at 37°C while shaking. These cultures were centrifuged for 5 min at 6,720 g and then resuspended in fresh LB. The donor and the recipient strains were mixed in a 1:1 ratio and incubated 4 h. Then, aliquots were plated on LB plates supplemented with appropriate antibiotics (with or without ALA). After overnight incubation, isolated colonies were collected and suspended in LB with appropriate antibiotics. Cultures of donors or recipients were used as negative controls.

### Collection and processing of wastewater samples

Samples were taken from the influence of two municipal WWTPs in Zealand, Eastern Denmark. Hillerød Centralrenseanlæg Syd, which is a WWTP that treats water of the city of Hillerød (1.3 × 10$^5$ person equivalent [PE]) and Lundtofte WWTP, which treats water of the Lyngby-Taarbæk Municipality (2.5 × 10$^4$ PE). Samples were collected in March 2022 using 24 h flow-proportional sampling in Hillerød WWTP (60) and grab sample collection in Lundtofte WWTP and then kept at 4°C until their use within 24 h.

Samples were centrifuged at 8,000 g for 1 h at 4°C, and then the supernatant was collected to remove big particles and most bacteria. To remove the remaining bacterial debris while retaining the viral fraction, the supernatant passed through a sterile 25-mm Whatman glass fiber membrane with a pore size of 0.7 µm (Sigma-Aldrich). These samples were collected in sterile 50-mL tubes and stored at 4°C.

### Phage isolation and purification

To remove the somatic phages, first, we inoculated the samples with *S. enterica* MHM112 without plasmids and incubated for 15 min. Then, samples were centrifuged and filtered to remove residual bacterial cells. To isolate plasmid-dependent phages, we used the DLA (61) with same bacterial strain *S. enterica* MHM112 but carrying plasmid pKM101. For this, bacterial strain was grown overnight in LB broth with ampicillin 100 µg mL$^{-1}$ at 25°C (120 rpm). Then, 10 mL of fresh LB was inoculated with 100 µL of the overnight culture and incubated for 120 min at 25°C under shaking at 120 rpm, to get mid-exponential

growing phase (~$5 \times 10^7$ CFU mL$^{-1}$). Aliquots of 100 µL of bacterial suspensions were mixed with 100 µL of filtered wastewater samples and 3-mL melted (50°C) soft-LB agar (0.7%) supplemented with CaCl$_2$ (final concentration 5 mM). After overnight incubation at 25°C, plaques were taken from each sampling location and harvested in 500-µL SM buffer (100-mM NaCl, 8-mM MgSO$_4$, 50-mM Tris-HCl, and pH 7.5). Then, the isolates were purified three times by DLA and sequential isolation. Subsequently, we checked if the isolates were plasmid-dependent, using the strain *S. enterica* MHM112 without plasmids by DLA. For further experiments, one phage of each sampling location was used (phage Hi226 from Hillerød WWTP and phage Lu221 from Lundtofte WWTP).

## Concentration of phage suspensions

Concentrated stocks of phages Lu221 and Hi226 were obtained using Amicon ultrafiltration membranes (100 kDa, MilliporeSigma). For that, phage suspensions were mixed with logarithmic growing cultures of *S. enterica* (pKM101) as described before. Then, phages were added at MOI 0.01 in 15-mL bacterial cultures. Samples were incubated overnight at 25°C under shaking at 120 rpm. After incubation, cultures were transferred to Amicon tubes and centrifuged at $3,000 \times g$ for 20 min at 4°C or until a remaining volume lower than 1 mL. Finally, the titer of the phage suspensions was determined by DLA as described before and expressed as plaque-forming units mL$^{-1}$ (PFU mL$^{-1}$).

## Resistance to RNAse and chloroform

To determine if the isolated phages were RNA phages, RNAse resistance was determined by DLA, adding RNAse to the bottom agar at 10 µg mL$^{-1}$. Their resistance to chloroform was determined by adding it at final concentration of 10 or 100 µL mL$^{-1}$ in 200 µL of phage suspensions. Mixtures were incubated 60 min at 25°C, and phage viability was determined by DLA. All assays were done by triplicate.

## Electron microscopy

Electron microscopy was performed to visualize the morphology of the isolated phages using an aliquot of 5 µL from pure suspensions with glow discharge 200 mesh copper-coated grids (Ted Palla, Inc.) using Leica Coater ACE 200 (30 s, 10 mA). Phages on the grids were incubated for 30 s before bloting off liquid using a Whatman filter paper. Samples were then fixed with 5 µL of glutaraldehyde, incubated for 10 s, and blot off excess liquid. Then, samples were stained with 3 µL of 2% uranyl acetate and incubated for 30 s. The transmission electron microscope used was CM100, with a LaB6 emitter. TEM images were taken with an Olympus Veleta camera and analyzed with ImageJ software.

## Adsorption assay

Mid-exponential growing phase cultures of *S. enterica* (pKM101) were inoculated with phage Lu221 or Hi226 at MOI 0.001 (62). Aliquots of 500 µL were taken (0, 15, 30, 45, 60, 75, 90, 105, and 120 min) and centrifuged 2 min at 14,100 g. The phage particles in the supernatants were enumerated by DLA in triplicates. Fractions of non-adsorbed and adsorbed phages at each time point were calculated. In addition, the adsorption rate constant was determined using k = -ln($P$/Po)/Nt, where $N$ is the bacterial density, Po and $P$ are the starting and ending phage titers, $k$ is the adsorption rate constant, and $t$ is the time in minutes over which adsorption occurs.

## One-step growth curve

The latent period and burst size of phages Lu221 and Hi226 were determined in a one-step growth curve experiment. For that, mid-exponential growing phase cultures of *S. enterica* (pKM101) were inoculated with each phage at MOI 1 (5 mL in LB broth at $5 \times 10^7$ CFU mL$^{-1}$). After incubation for 15 min at 25°C, 1 mL was centrifuged 3 min at 8,000 $\times$ g, and the supernatant was discarded. The pellet was resuspended in 100-mL fresh

LB and kept at room temperature without shaking. Aliquots of 100 µL were taken (at 0, 10, 20, 30, 40, 50, and 60 min), and phage titer was determined by DLA in triplicates. Burst size was calculated by dividing the phage titers at plateau phase by the number of infected bacterial cells.

## Growth inhibition assay

Mid-exponential growing phase cultures of *S. enterica* with or without plasmid pKM101 were inoculated with phages Lu221 or Hi226 at MOI 10 in duplicate flasks of 250 mL containing 100 mL of LB broth and incubated at 25°C without shaking. Samples of 1 mL were taken every 30 min until 210 min. The absorbance was measured by spectrophotometry at 600 nm. Controls without phages were included.

## Host range

We explored our isolate infectivity against diverse bacteria (*S. enterica*, *E. coli*, *Kluyvera* sp., *Enterobacter* sp., and *P. putida*) carrying one conjugative plasmid from different Inc classes (IncF, IncH, IncI, IncN, IncP, IncW, and IncX). All the plasmids carried resistance determinants (Table S1). For each bacterial strain and phage isolate, we incubated a mixture of overnight culture and phage at different concentrations (four decimal dilutions) for 20 min before pipetting a drop (20 µL) on LB agar in triplicates (63). The appearance of plaques in the lawn after overnight incubation at 25°C was indicative of the phage ability to cause lytic infection (Fig. S1). If too many plaques prevented counting at the fourth decimal dilution, experiments with the next two dilutions were performed.

## DNA extraction and sequencing

DNA extraction was carried out using PureLink Viral RNA/DNA Mini Kit (Invitrogen) following the user manual. The starting material was 0.2 mL of phage stocks. The purified DNA was then visualized by agarose gel electrophoresis, and its concentration was quantified using the Qbit dsDNA kit. The DNA was sequenced by Copenhagen University using an Illumina HiSeq 2500 platform.

## Genome analysis

Raw sequences of phages Lu221 and Hi226 were quality-checked using FastQC version 0.11.8 and then trimmed and assembled in a single contig with the online tool PATRIC 3.6.12 (64, 65). A Megablast was run to determine their similarity with other described genomes, using default parameters and considering only complete genomes (66). Then, these similar phages sequences (33 complete genomes) were used as references in the further analysis. Pairwise comparisons of the nucleotide sequences were conducted using the Genome BLAST Distance Phylogeny (GBDP) method (67) with settings recommended for prokaryotic viruses. To classify phages Lu221 and Hi226 and construct a phylogenetic tree, we used the online resource VirusTaxo (68). In addition, we used the Virus Classification and Tree Building Online Resource web service (https://victor.dsmz.de) (69) and VIRIDIC (70).

## Annotation

The annotation of the genome of phage Lu221 was done using Phage Commander application for rapid identification of bacteriophage genes using multiple gene identification programs (71, 72). Phage Commander runs a bacteriophage genome sequence through nine gene identification programs (and an additional program for identification of tRNAs), such as RAST (73, 74) and GeneMarks (75). tRNA genes were searched using ARAGORN (76). To infer functions of the predicted genes, BLAST (blastx) was used, with parameter options set to default values, and the $E$-value threshold set to $10^{-4}$ (77).

## ACKNOWLEDGMENTS

This project has received funding from the European Union's Horizon 2020 research and innovation programme under the Marie Sklodowska-Curie grant agreement No 101026675 and was supported by a research grant "P-PhanFARE (23046)" from VILLUM FONDEN.

We appreciate the skillful contributions of Marco Cerqueda, Zhiming He, and Rosa Lusetti to the laboratory experiments and the assistance of the staff at the wastewater treatment plants Mølleåværket (Lyngby-Taarbæk Forsyning A/S) and Hillerød Centralrenseanlæg Syd (Hillerød Forsyning) for sample collection.

## AUTHOR AFFILIATIONS

[1]Department of Environmental Engineering and Resource Engineering, Technical University of Denmark, Kongens Lyngby, Denmark

[2]Laboratorio de Investigación de Agentes Antibacterianos, Departamento de Microbiología, Facultad de Ciencias Biológicas, Universidad de Concepción, Concepción, Chile

[3]Instituto de Ciencias Naturales, Facultad de Medicina Veterinaria y Agronomía, Universidad de las Américas, Concepción, Chile

[4]Department of Veterinary and Animal Sciences, University of Copenhagen, København, Denmark

## AUTHOR ORCIDs

Boris Parra http://orcid.org/0000-0002-8416-3514
Arnaud Dechesne http://orcid.org/0000-0002-6638-2158

## FUNDING

| Funder | Grant(s) | Author(s) |
| --- | --- | --- |
| EC \| Horizon Europe \| Excellent Science \| HORIZON EUROPE Marie Sklodowska-Curie Actions (MSCA) | 101026675 | Boris Parra |
| Villum Fonden (Villum Foundation) | 230416 | Arnaud Dechesne |

## AUTHOR CONTRIBUTIONS

Boris Parra, Conceptualization, Data curation, Formal analysis, Funding acquisition, Investigation, Methodology, Project administration, Resources, Supervision, Validation, Visualization, Writing – original draft, Writing – review and editing | Bastiaan Cockx, Investigation, Methodology, Writing – review and editing | Veronika T. Lutz, Data curation, Formal analysis, Investigation, Methodology, Writing – review and editing | Lone Brøndsted, Formal analysis, Investigation, Methodology, Supervision, Writing – review and editing | Barth F. Smets, Conceptualization, Formal analysis, Writing – review and editing | Arnaud Dechesne, Conceptualization, Formal analysis, Funding acquisition, Investigation, Methodology, Project administration, Resources, Supervision, Validation, Writing – review and editing

## DATA AVAILABILITY

The genome sequences of phages Lu221 and Hi226 have been deposited in GenBank under accession no. OQ829281 and OR053651, respectively.

## ADDITIONAL FILES

The following material is available online.

### Supplemental Material

**Supplemental material (Spectrum02537-S0001.docx).** Tables S1 to S3; Fig. S1.

## Open Peer Review

**PEER REVIEW HISTORY (review-history.pdf).** An accounting of the reviewer comments and feedback.

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
