## [Reviewer comments · Microbiology Spectrum]

Microbiology Spectrum

Isolation and Characterization of Novel Plasmid-dependent Phages Infecting Bacteria Carrying Diverse Conjugative Plasmids

Boris Parra, Bastiaan Cockx, Veronika T. Lutz, Lone Brøndsted, Barth F. Smets, and Arnaud Dechesne

Corresponding Author(s): Arnaud Dechesne, Danmarks Tekniske Universitet

Review Timeline:

Submission Date:	June 19, 2023
Editorial Decision:	September 16, 2023
Revision Received:	October 24, 2023
Accepted:	November 12, 2023

Editor: Daria Van Tyne

Reviewer(s): Disclosure of reviewer identity is with reference to reviewer comments included in decision letter(s). The following individuals involved in review of your submission have agreed to reveal their identity: Matti Jalasvuori (Reviewer #1)

Transaction Report:

DOI: <https://doi.org/10.1128/spectrum.02537-23>

September 16, 2023

Dr. Arnaud Dechesne
Danmarks Tekniske Universitet
Department of Environmental Engineering and Resource Engineering
Kongens Lyngby
Denmark

Re: Spectrum02537-23 (Isolation and Characterization of Novel Plasmid-specific Phages Infecting Bacteria Carrying Diverse Conjugative Plasmids)

Dear Dr. Arnaud Dechesne:

Thank you for submitting your manuscript to Microbiology Spectrum. Your manuscript was reviewed by two experts, and I would now like you to revise your study in line with their feedback.

Link Not Available

Sincerely,

Daria Van Tyne

Journals Department
Editor comments:

Please add a "Data Availability" section to the Methods that includes accession information for WGS and other data generated in the study.

Reviewer comments:

Reviewer #1 (Comments for the Author):

Line 131: The second "then" could be removed.

Line 138: The line could specify "what length" is being measured.

Lines 239 and 241: The description of the burst size could be written more clearly, as the PFU per infected cell is much lower, and it is difficult to estimate the burst size as the number of infected cells is unknown.

Overall the study is sound and the characterized phages are intriguing additions to the plasmid-dependent group. In the future, it would be nice to identify common elements in these plasmids to pinpoint potential factors that provide the broad host range of the phages. Also, the potential phenotypic changes in the bacterial populations exposed to phage predation could be studied.

Reviewer #2 (Comments for the Author):

The authors present the discovery of two new Caudoviricetes, named Lu221 and Hi226. They proceed to characterize the two phages and report that they are plasmid-dependent. From this the authors claim that these are promising as a solution for antimicrobial resistance.

In practice, the two phages are extremely similar, and on all reported tests either they are the same or error ranges are not reported, making it impossible to evaluate whether there is anything beyond experimental noise differentiating the two phages.

Overall, while the experiments reported appear largely solid, the conclusions drawn are not very well supported. In particular, the central claim that these phages have the "broadest coverage in terms of Inc groups" is not well supported, and in its current form is not terribly meaningful. This is because, as the authors note, the correspondence between Inc-type and pilus type is not strong, and so the same pilus type can be carried on plasmids of many Inc-types, leading to the appearance of a broad host range. Even if this were a meaningful distinction for a phage, it is not clear the reason other phages are not similar isn't an artifact of this simply not being tested for other phages because it is orthogonal to what they actually infect.

Further, as no selection against resistance beyond what is already cited by this paper is shown, it seems overly broad to claim that this has new potential against resistance without evidence.

Major comments:

-Conjugative plasmid incompatibility type is not mechanistically related to plasmid-encoded conjugation machinery, so though it is pragmatic to classify phages based on what Inc group plasmids they infect, analysis of plasmid conjugation genes is necessary to interpret claims of broad host range plasmid dependent phages. How do the authors know that the conjugal pilus of their IncF, IncH, IncI, IncN, IncP, IncW and IncX representative plasmids are not related? Some genomic comparisons of the relatedness of the plasmids used are needed. For example, the authors could include a CLINKER plot of all the plasmids used in the study.

-In particular, if the authors wish to claim (per line 299) that these two phages have "the widest host range in terms of plasmid diversity" of all known phages, this is a critical point, as there does not appear to be a reason given why diversity in terms of replication system is a meaningful figure of merit for phages that target the conjugative pilus independent of replication system. (As a thought exercise, what would it do to the notion of diversity if the pilus from the tested phages was expressed chromosomally and permitted phage infection?)

-Whenever rates are reported from plaque-counting data, for example the adsorption rates reported on lines 239-240, error ranges should be reported. If the authors are unfamiliar, this is generally at the inverse of the square root of the total number of plaques counted between replicates. This is critical, as data derived from a small number of plaques could misleadingly give a difference in adsorption or plaquing efficiency where in fact none exists. The authors must report this binomial error for all numbers in the paper derived from counting plaques or colonies.

-The host range data is extremely difficult to interpret in table form (Table 4). It would be much better represented as a heatmap. Do the PFU/mL value in represent a single measurement? Plaque assays can be noisy and therefore I would advise the authors to show average values of triplicate assays. Additionally, it would be nice if the authors could show example plaque assays for some of the more surprising differences, e.g. plaquing with the pKJK5 plasmid only in Salmonella and not E. coli. It would be useful to know if there was any clearing at all of the lawn at high concentrations of phage, even if there were no plaques.

-The finding that Lu221 and Hi226 can plaque on IncP plasmids in Salmonella but not E. coli is very strange (line 269). The authors very briefly speculate that this could be due to differential expression of plasmid pilus genes in different bacteria. But this is extremely easy to test. Could the authors add an adsorption assay for one representative IncP plasmid in E. coli and Salmonella, this would indicate whether the pilus is actually being expressed in their E. coli IncP strains? Alternatively, a conjugation efficiency assay from the E. coli IncP strains could also demonstrate that the pilus is intact.

-Phylogenetic tree in Figure 4 is misleading because it appears to be based on mostly very small regions of identity (<1% query

cover) shown in Table 3. Authors should check what that region is, and if it is the same between all the phages. If it is a structural protein (as is suggested at line 206), then that indicates that a better approach would be to extract only the conserved structural protein (i.e. by running BLASTP on just this protein from Lu221 or Hi226, and building a phylogenetic tree from these results (ignoring results that have very low query cover).

Minor comments:

-The authors repeatedly use non-standard terminology. In particular, the use of "somatic" to mean bacteria without a conjugative pilus, while likely meant as analogous to metazoan germline and somatic cells, is both nonstandard and an incorrect metaphor. Moreover this distinction misleads from the important difference, which is whether the cell expresses a pilus that the phage can infect or not.

-Further, even in the title "plasmid specific phage" is used -- while technically all these phages are pilus-dependent, the typical term in the field is "plasmid dependent". I would suggest the authors use standard terminology such as "plasmid negative" to aid readers in the field.

-Line 170, the clinker figure is not a genome alignment but an alignment of all annotated protein sequences. This should be clarified in the legend, and it would be helpful in the key said "amino acid identity (%)" to avoid confusion

-Text at line 182 is very vague. Which are the "additional genes" and "other genes". It would be helpful if the authors could add insight into the functional identity of the genes which are different between the phages.

-Lines 189-215: The grounds the authors are using to classify phages as "related" are unclear to me. Table 3 appears to show a BLAST output where most the hits are to extremely small regions of the queries (<1% query cover in most cases). This means a <700 bp piece of Lu221 or Hi226 is hitting these phage genomes. I would not consider phages that match less than 10% of the genome to related unless there is other evidence, such a conserved core structural protein, or conserved genome synteny.

-The strain of Salmonella used by the authors to isolate phages appears to be cured of the IncF-like Salmonella virulence plasmid. This information might be worth noting in the methods, as wildtype Salmonella strains may not be permissive to all the plasmids the authors have used in this study.

-Methods section starting at line 342 suggests that all plaque assays were conducted at 25 degrees Celsius. Was this just for Salmonella or for all hosts used. It is not clear in the methods, and this is an important detail to include as typically all work with enteric bacteria is conducted at their optimal growth temperatures (30-37 degrees Celsius).

-Figure S1 is uninformative. What is the figure showing? Authors should fully describe the figure in the legend, and indicate why this method alone requires visual documentation in the supplementary figures.

-Could the authors give more information about the strains and plasmids they are using. Typically, a table is included in the supplementary including every strain used, the genome accession if available, and growth conditions. Plasmids should also be included, along with a genome accession number/ key reference or source (this could be merged with the current Table S1).

Staff Comments:

Preparing Revision Guidelines

Please return the manuscript within 60 days; if you cannot complete the modification within this time period, please contact me. If you do not wish to modify the manuscript and prefer to submit it to another journal, please notify me of your decision immediately so that the manuscript may be formally withdrawn from consideration by Microbiology Spectrum.

Editor comments

Please add a "Data Availability" section to the Methods that includes accession information for WGS and other data generated in the study.

Answer: "Data availability was incorporated" (please see lines 405-407 in the new version).

Reviewers comments

Reviewer #1

Line 131: The second "then" could be removed.

Answer: The second "then" was removed (please see line 109 in the new version).

Line 138: The line could specify "what length" is being measured.

Answer: We changed: "Phage Lu221 has a length of 152.37 ± 10.39 nm and a width of 69.00 ± 8.58 nm, while phage Hi226 has a length of 150.96 ± 10.18 nm and a width of 70.53 ± 9.01 nm"

To: "The head of phage Hi226 has a length of 152 ± 10 nm and a width of 69 ± 8 nm, while phage Lu221 has a head length of 150 ± 10 nm and a width of 70 ± 9 nm".

Please see line 116-118 in the new version.

Lines 239 and 241: The description of the burst size could be written more clearly, as the PFU per infected cell is much lower, and it is difficult to estimate the burst size as the number of infected cells is unknown.

Answer: The initial number of phages and the number of infected cells is known and was included in the methods section of the new version (please see line 358). In addition, we realized that there was an error in the burst size of phage Hi226 – we had written 10^7 , but it should read 107 PFU/infected cell (it was expressed correctly in the Table 2). This was corrected in the manuscript (please see line 204 in the new version).

Overall the study is sound and the characterized phages are intriguing additions to the plasmid-dependent group. In the future, it would be nice to identify common elements in these plasmids to pinpoint potential factors that provide the broad host range of the phages. Also, the potential phenotypic changes in the bacterial populations exposed to phage predation could be studied.

Answer: We thank the reviewer for his supportive comments. We agree that having more sequences of plasmid-dependent phages will allow to identify what controls the specificity of these phages. This is beyond the scope of this paper, which focuses on basic description of the novel isolated plasmid-dependent phages.

Reviewer #2

The authors present the discovery of two new Caudoviricetes, named Lu221 and Hi226. They proceed to characterize the two phages and report that they are plasmid-dependent. From this the authors claim that these are promising as a solution for antimicrobial resistance.

In practice, the two phages are extremely similar, and on all reported tests either they are the same or error ranges are not reported, making it impossible to evaluate whether there is anything beyond experimental noise differentiating the two phages.

Overall, while the experiments reported appear largely solid, the conclusions drawn are not very well supported. In particular, the central claim that these phages have the "broadest coverage in terms of Inc groups" is not well supported, and in its current form is not terribly meaningful. This is because, as the authors note, the correspondence between Inc-type and pilus type is not strong, and so the same pilus type can be carried on plasmids of many Inc-types, leading to the appearance of a broad host range. Even if this were a meaningful distinction for a phage, it is not clear the reason other phages are not similar isn't an artifact of this simply not being tested for other phages because it is orthogonal to what they actually infect.

Further, as no selection against resistance beyond what is already cited by this paper is shown, it seems overly broad to claim that this has new potential against resistance without evidence.

Answer: We agree that "the correspondence between Inc-type and pilus type is not strong, and so the same pilus type can be carried on plasmids of many Inc-types, leading to the appearance of a broad host range". Therefore, we decided to tone down our claims based on Inc groups and focus on host range in terms of bacterial strains.

We also agree that our claims on the potential against antimicrobial resistance were insufficiently supported. We changed the wording and make it clear that our phages infected bacteria carrying twelve out against fifteen conjugative plasmids carrying resistance determinants. Our previous table S1, listing plasmids was incomplete, but now is complete (please see the new version in lines 679-682).

We edited the manuscript to make these points clear:

In the abstract, the phrase: "Their host range is extremely broad and covers conjugative plasmids from several Inc groups: IncF, IncH, IncI, IncN, IncP, IncW and IncX, making them the plasmid-specific phages with the broadest coverage in terms of Inc groups" was changed to: "They showed broad host range infecting *E. coli*, *S. enterica*, *Kluyvera sp.* and *Enterobacter sp.* carrying conjugative plasmids. They recognize plasmid-encoded receptors from 12 out of 15 tested plasmids, all of them carrying resistance determinants". Please see lines 34-36 in the new version.

Please see the new "Importance" section (lines 46-54). We changed:

"This work was undertaken because plasmid-specific phages can reduce the prevalence of conjugative plasmids and can be leveraged to prevent the acquisition and dissemination of ARGs by bacteria. The two novel phages described in this study, Lu221 and Hi226, both have an unusually broad host range; broader than previously described for plasmid-specific phages and including bacteria carrying plasmids of the most common Inc plasmid types among Gram-negative pathogens: IncF, IncH, IncI, IncN, IncP, IncW and IncX. Such plasmids are known to carry ARGs and, if not controlled, to rapidly spread within and across

species, facilitating the rise of multidrug-resistant, extremely drug resistant and pan drug-resistant pathogens. Therefore, the novel isolates phage Lu221 and Hi226 have a great potential to target bacteria carrying conjugative plasmids and mitigate the antimicrobial resistance crisis”.

To: “This work was undertaken because plasmid-dependent phages can reduce the prevalence of conjugative plasmids and can be leveraged to prevent the acquisition and dissemination of ARGs by bacteria. The two novel phages described in this study, Lu221 and Hi226, can infect *E. coli*, *S. enterica*, *Kluyvera sp.* and *Enterobacter sp.* carrying conjugative plasmids. This was verified with plasmids carrying resistance determinants and belonging to the most common plasmid families among Gram-negative pathogens. Therefore, the newly isolated phages could have the potential to help control the spread of ARGs and thus help combat the antimicrobial resistance crisis”.

Please see the new conclusion (lines 269-275). We changed:

“In summary, in this work, we describe two novel lytic plasmid-specific phages isolated from wastewater, capable of infecting diverse bacteria carrying conjugative plasmids from several Inc classes. They have the widest host range in terms of plasmid diversity of all known plasmid-specific phages, which, together with other characteristics such as their burst size, make them a potential biotechnological tool to combat the AMR crisis by reducing the spread of ARGs in Gram-negative bacteria”.

To: “In conclusion, in this work we describe two novel lytic plasmid-dependent phages isolated from wastewater, Lu221 and Hi226. They showed a broad host range in terms of bacterial strains and were able to recognize twelve conjugative plasmids carrying resistance determinants and belonging to the most common plasmid families among Gram-negative pathogens. The characteristics of phages Lu221 and Hi226 make them potential agents to help combat the AMR crisis by reducing the spread of ARGs commonly present on conjugative plasmids.”

Major comments:

-Conjugative plasmid incompatibility type is not mechanistically related to plasmid-encoded conjugation machinery, so though it is pragmatic to classify phages based on what Inc group plasmids they infect, analysis of plasmid conjugation genes is necessary to interpret claims of broad host range plasmid dependent phages. How do the authors know that the conjugal pilus of their IncF, IncH, IncI, IncN, IncP, IncW and IncX representative plasmids are not related? Some genomic comparisons of the relatedness of the plasmids used are needed. For example, the authors could include a CLINKER plot of all the plasmids used in the study.

Answer: The reviewer is correct that differences in plasmid incompatibility groups are not expected to necessarily translate in difference in mating pair apparatus (which controls infectivity by plasmid-dependent phages). Indeed, incompatibility and mating pair formation are governed by different sets of plasmid genes: genes related to replication and segregation for one, and to biogenesis of the pilus and the other proteins present in the T4SS for the other. Whereas the pilus gene is not the only determinant for plasmid-dependent infection (e.g. for phage PRD1, both the Tra2 core region (*trbB, C, D, E, F, G, H, I, J, K, L*) and *traF* are required (Daugelavičius et al 1997, Haase et al., 1995)), it is not currently clear which gene homologues are relevant to compare or contrast our model plasmids. Instead, we find it

justified to use the Inc classification because there have been many studies on the relation between Inc grouping and plasmid-dependent infectivity (Bradley et al., 1981; Olsen et al., 1974; Bamford et al., 1995). It is established that some incompatibility groups have similar infectivity patterns because they have similarity in their pilus and/or other key components of the T4SS (Frost, 1993).

In response to the reviewer's comment, we decreased the emphasis on host range in terms of Inc families and provided more on host range in terms of bacterial strains. We edited the “Host range” section on Results and Discussion to make this point clear. We changed: “We tested the ability of phages Lu221 and Hi226 to infect diverse bacteria, such as *E. coli*, *S. enterica* or *Pseudomonas putida*, *Kluyvera* sp. and *Enterobacter* sp., carrying plasmids from seven Inc families, IncH, IncI, IncN, IncP, IncW and IncX (Table 4). Our result revealed the very broad plasmid infection range of both phages. Indeed, they are able to infect bacteria carrying plasmids from all of these Inc families. So far, the plasmid-specific phage with the widest host range is phage X, which is an ssDNA filamentous phage able to infect IncI, IncM, IncN, IncP, IncW and IncX (Frost, 1993). Therefore, to the best of our knowledge, phages Lu221 and Hi226 have the broader host range in terms of Inc plasmid classes among all described plasmid-specific phages”.

To: “We tested the ability of phages Lu221 and Hi226 to infect diverse bacteria (Fig. 6). Our results revealed the broad host range of isolated phages, because they infected plasmid-containing *E. coli*, *S. enterica*, *Kluyvera* sp. and *Enterobacter* sp. They were unable to infect *P. putida*, which suggests a specificity towards Enterobacteriaceae. In this regard, other plasmid-dependent phages have been showed broad host range; for instance, PRD1 can infect *Salmonella*, *Pseudomonas*, *Escherichia*, *Proteus*, *Vibrio*, *Acinetobacter*, and *Serratia* strains carrying conjugative plasmids (Olsen et al., 1974; Bamford et al., 1995). In terms of plasmid diversity, they are able to infect Enterobacteriaceae carrying twelve of the fifteen plasmids carrying resistance determinants tested, spanning seven Inc families, IncH, IncI, IncN, IncP, IncW and IncX. Another plasmid-dependent phage able to infect a similarly diverse range of plasmids is phage X, which is an ssDNA filamentous phage (Bradley et al., 1981). However, this phage was not taxonomically classified, and its genome sequence is not available (Frost, 1993).

Please, see lines 217-228 in the new version.

-In particular, if the authors wish to claim (per line 299) that these two phages have “the widest host range in terms of plasmid diversity” of all known phages, this is a critical point, as there does not appear to be a reason given why diversity in terms of replication system is a meaningful figure of merit for phages that target the conjugative pilus independent of replication system. (As a thought exercise, what would it do to the notion of diversity if the pilus from the tested phages was expressed chromosomally and permitted phage infection?)

Answer: The reviewer is correct, and we appreciate his comments. We edited our manuscript to focus the host range determination in terms of bacterial strains rather than Inc families.

We changed: “the widest host range in terms of plasmid diversity” to: “They showed a broad host range in terms of bacterial strains and were able to recognize twelve conjugative plasmids carrying resistance determinants and belong to the most common plasmid families among Gram-negative pathogens”. Please see lines 270-273 in the new version.

-Whenever rates are reported from plaque-counting data, for example the adsorption rates reported on lines 239-240, error ranges should be reported. If the authors are unfamiliar, this is generally at the inverse of the square root of the total number of plaques counted between replicates. This is critical, as data derived from a small number of plaques could misleadingly give a difference in adsorption or plaquing efficiency where in fact none exists. The authors must report this binomial error for all numbers in the paper derived from counting plaques or colonies.

Answer: The reviewer is correct, and we appreciate his comments. Error ranges were reported in the new version, please see lines 204 and 205.

-The host range data is extremely difficult to interpret in table form (Table 4). It would be much better represented as a heatmap. Do the PFU/mL value in represent a single measurement? Plaque assays can be noisy and therefore I would advise the authors to show average values of triplicate assays. Additionally, it would be nice if the authors could show example plaque assays for some of the more surprising differences, e.g. plaquing with the pKJK5 plasmid only in *Salmonella* and not *E. coli*. It would be useful to know if there was any clearing at all of the lawn at high concentrations of phage, even if there were no plaques.

Answer: We appreciate the reviewer comments. Host range data were reported as heatmap. Please see new Figure 6 (line 229).

The PFU/mL values represent average values of triplicates measurements (please see line 376 in methodology section). Error ranges were reported - Please see Table S3 in the new version (line 694).

Even at higher concentration, no clearing was observed in the lawn of negative strains. This was included in the text, please see lines 232-233.

-The finding that Lu221 and Hi226 can plaque on IncP plasmids in *Salmonella* but not *E. coli* is very strange (line 269). The authors very briefly speculate that this could be due to differential expression of plasmid pilus genes in different bacteria. But this is extremely easy to test. Could the authors add an adsorption assay for one representative IncP plasmid in *E. coli* and *Salmonella*, this would indicate whether the pilus is actually being expressed in their *E. coli* IncP strains? Alternatively, a conjugation efficiency assay from the *E. coli* IncP strains could also demonstrate that the pilus is intact.

Answer: Conjugation efficiency assay from *E. coli* (RP4) was tested, demonstrating that this strain is able to transfer the plasmid, and therefore the pilus is likely intact - this was incorporated in the manuscript (please see lines 242-243).

The differential infectivity of a plasmid-dependent phage across different bacteria carrying the same conjugative plasmid as we report here has been reported previously. This differential behavior can be due to many reasons that we listed, supported by references, in the manuscript: a) different expression of *mpf* genes, b) a non-plasmid coded structure stabilizing phage attachment, or c) bacteria differ in their defense mechanisms against phages (antiphage systems). We edited the manuscript to make this point clear (please see lines 244-247).

-Phylogenetic tree in Figure 4 is misleading because it appears to be based on mostly very small regions of identity (<1% query cover) shown in Table 3. Authors should check what

that region is, and if it is the same between all the phages. If it is a structural protein (as is suggested at line 206), then that indicates that a better approach would be to extract only the conserved structural protein (i.e. by running BLASTP on just this protein from Lu221 or Hi226, and building a phylogenetic tree from these results (ignoring results that have very low query cover)).

Answer: In our opinion, the BLAST analysis between our isolates and the other phages reported in Table 3 cannot be used to evaluate the reliability of the phylogenetic tree in Figure 4. Indeed, this tree is inferred from full pairwise comparisons of whole genomes using Virus Classification and Tree Building Online Resource (VICTOR), which is designed to compare even distant prokaryotic viruses. VICTOR tree is not based on a small piece of sequence that would be homologous across all genomes: it measures pairwise distance between all phages from their genome sequences. Some phages are quite distant from our isolates sequences, but others are closer, so we have a meaningful distribution of pairwise distances in the distance matrix (as proven by the high bootstrap values on most of the main branches). We refer the reviewer to the manuscript that describes VICTOR, which present a phage phylogenetic tree across a dozen of families [VICTOR: genome-based phylogeny and classification of prokaryotic viruses - PubMed \(nih.gov\)](https://pubmed.ncbi.nlm.nih.gov/31111111/)

To focus Table 3 on the most relevant comparisons, we have edited it considering only phages in the same family (please see lines 674-676).

Minor comments:

-The authors repeatedly use non-standard terminology. In particular, the use of "somatic" to mean bacteria without a conjugative pilus, while likely meant as analogous to metazoan germline and somatic cells, is both nonstandard and an incorrect metaphor. Moreover, this distinction misleads from the important difference, which is whether the cell expresses a pilus that the phage can infect or not. Further, even in the title "plasmid specific phage" is used -- while technically all these phages are pilus-dependent, the typical term in the field is "plasmid dependent". I would suggest the authors use standard terminology such as "plasmid negative" to aid readers in the field.

Answer: We agree that the term "somatic" is not precise for bacteria without a conjugative pilus and therefore, we removed it from the manuscript. However, we wish to retain our use of 'somatic' for non-plasmid-dependent phages as there are many examples in the literature in the same way as we do, especially in the field of coliphage detection for water quality assessment, e.g:

<https://academic.oup.com/femsle/article/363/17/fnw180/2197723>;

<https://ami-journals.onlinelibrary.wiley.com/doi/full/10.1111/j.1365-2672.2008.03957.x>;

<https://www.frontiersin.org/articles/10.3389/fmicb.2022.941532/full>)

It is also used in a standard method (https://www.epa.gov/sites/default/files/2015-12/documents/method_1601_2001.pdf).

We agree with the reviewer that while plasmid-specific phage and plasmid-dependent coexist in the literature, the latter is maybe more exact, and we thus changed the term "plasmid-specific" to "plasmid-dependent" in our manuscript. For instance, please see lines 1, 24, 26, 29).

-Line 170, the clinker figure is not a genome alignment but an alignment of all annotated protein sequences. This should be clarified in the legend, and it would be helpful in the key said "amino acid identity (%)" to avoid confusion.

Answer: the clinker figure legend was clarified (please see lines 137-138).

-Text at line 182 is very vague. Which are the "additional genes" and "other genes". It would be helpful if the authors could add insight into the functional identity of the genes which are different between the phages.

Answer: phages Lu221 and Hi226 have lower GC content and shorter genomes compared to their close relatives. The genes that are differentially present encode hypothetical proteins with unknown functions (please see lines 149-150).

-Lines 189-215: The grounds the authors are using to classify phages as "related" are unclear to me. Table 3 appears to show a BLAST output where most the hits are to extremely small regions of the queries (<1% query cover in most cases). This means a <700 bp piece of Lu221 or Hi226 is hitting these phage genomes. I would not consider phages that match less than 10% of the genome to related unless there is other evidence, such a conserved core structural protein, or conserved genome synteny.

Answer: Table 3, was edited considering only phages in the same family (please see lines 674-676).

-The strain of *Salmonella* used by the authors to isolate phages appears to be cured of the IncF-like *Salmonella* virulence plasmid. This information might be worth noting in the methods, as wildtype *Salmonella* strains may not be permissive to all the plasmids the authors have used in this study.

Answer: the strain of *Salmonella* used to isolate phages was cured of the IncF-like plasmid. It was added to methods (please see line 278).

-Methods section starting at line 342 suggests that all plaque assays were conducted at 25 degrees Celsius. Was this just for *Salmonella* or for all hosts used. It is not clear in the methods, and this is an important detail to include as typically all work with enteric bacteria is conducted at their optimal growth temperatures (30-37 degrees Celsius).

Answer: all plaque assays were conducted at 25 degrees Celsius. It was clarified in the methods (please see line 377).

-Figure S1 is uninformative. What is the figure showing? Authors should fully describe the figure in the legend, and indicate why this method alone requires visual documentation in the supplementary figures.

Answer: Figure legend in S1 was completed (please see lines 700-704).

-Could the authors give more information about the strains and plasmids they are using. Typically, a table is included in the supplementary including every strain used, the genome

accession if available, and growth conditions. Plasmids should also be included, along with a genome accession number/ key reference or source (this could be merged with the current Table S1).

Answer: Table S1 was completed with the used strains and plasmids. Please see new version in lines 679-682).

Re: Spectrum02537-23R1 (Isolation and Characterization of Novel Plasmid-dependent Phages Infecting Bacteria Carrying Diverse Conjugative Plasmids)

Dear Dr. Arnaud Dechesne:

Your manuscript has been accepted, and I am forwarding it to the ASM production staff for publication. Your paper will first be checked to make sure all elements meet the technical requirements. ASM staff will contact you if anything needs to be revised before copyediting and production can begin. Otherwise, you will be notified when your proofs are ready to be viewed.

Sincerely,
Daria Van Tyne
Editor
Microbiology Spectrum